# Application of the IDEAS Framework in Adapting a Web-Based Physical Activity Intervention for Young Adult College Students

**DOI:** 10.3390/healthcare10040700

**Published:** 2022-04-09

**Authors:** Kimberly R. Hartson, Lindsay J. Della, Kristi M. King, Sam Liu, Paige N. Newquist, Ryan E. Rhodes

**Affiliations:** 1School of Nursing, University of Louisville, Louisville, KY 40202, USA; paige.napier@louisville.edu; 2Department of Communication, University of Louisville, Louisville, KY 40292, USA; lindsay.della@louisville.edu; 3Department of Health and Sport Sciences, University of Louisville, Louisville, KY 40292, USA; kristi.king@louisville.edu; 4School of Exercise Science, Physical and Health Education, University of Victoria, Victoria, BC V8W 2Y2, Canada; samliu@uvic.ca (S.L.); rhodes@uvic.ca (R.E.R.)

**Keywords:** IDEAS framework, physical activity, e-health, young adults

## Abstract

User-centered developmental processes are critical to ensuring acceptability of e-health behavioral interventions, and yet physical activity research continues to be inundated with top-down developmental approaches. The IDEAS (Integrate, Design, Assess, and Share) framework outlines a user-centered process for development of e-health interventions. The purpose of this manuscript is to describe the application of the IDEAS framework in adapting a web-based physical activity intervention for young adult college students. Steps 1–3 emphasized *integrating insights from users and theory* and Steps 4–7 focused on *iterative and rapid design with user feedback*. Data were collected via repeat qualitative interviews with young adult college students (*N* = 7). Resulting qualitative metathemes were engagement, accountability, and cultural fit. Therefore, intervention modifications focused on strategies to foster ongoing engagement with the program (e.g., increase interactivity), support personal and social accountability (e.g., private social media group), and provide a cultural fit within the college lifestyle (e.g., images relevant to student life). The resulting web-based intervention included eight weekly lessons, an expanded resource library, “how-to” videos, step and goal trackers, and a private social media group to be led by a wellness coach. In conclusion, the IDEAS framework guided an efficient, user-centered adaptation process that integrated empirical evidence and behavior change theory with user preferences and feedback. Furthermore, the process allowed us to address barriers to acceptability during the design and build stages rather than at later stages of pilot and efficacy testing.

## 1. Introduction

There is a critical need for the development of effective and scalable physical activity interventions designed for young adults. Evidence suggests that physical activity declines as adolescents transition through college and young adulthood [1], with less than half of college students achieving the recommended 150 min of moderate-intensity, or 75 min of vigorous-intensity, aerobic activity per week [2]. Even fewer young adults, 34%, meet activity guidelines when strength training on two or more days per week is included in the recommendation [3]. These statistics are concerning because insufficient physical activity can negatively affect several aspects of health and increase risk of chronic illness later in life [4].

Electronic health (e-health) can be an effective modality for delivering health behavior interventions, such as those designed to increase physical activity [5], and is a logical approach for physical activity interventions with young adults who report frequent use of technology in their daily lives [6]. However, the acceptability and usefulness of an e-health intervention with one population does not guarantee similar results with a different population. For example, e-health interventions developed for older audiences may not meet the preferences and expectations of young adults because young adults report different technology usage patterns than middle and older adults, including higher use of social media and brief video- and photo-based platforms such as Instagram, YouTube, and TikTok [6]. There is little empirical evidence regarding the desired features of e-health physical activity programs among young adult college student populations (e.g., Ding [7]; Middelweerd et al. [8]; Yan et al. [9]). As such, the use of user-centered, strategic frameworks to guide the development of behavioral interventions is recommended to ensure an understanding of factors that could affect acceptability of the intervention by a priority population [10]. 

Despite this recommendation, physical activity experimental research continues to be inundated with top-down developmental approaches that lack a user-centered focus [11]. Such top-down approaches can result in interventions that ignore participant needs and preferences and may be ineffective [11]. Few theoretical frameworks provide clear guidance on how to incorporate a user-centered approach into the process of developing an e-health behavior change intervention, a process that can quickly become complex as researchers must navigate the integration of behavior change theory and techniques, empirical evidence, user values and preferences, technology functionality, and available resources [12]. The IDEAS (Integrate, Design, Assess, and Share) framework (Figure 1) is one such framework created for the development of innovative and effective digital health behavior change interventions [13]. It incorporates iterative, user-centered design thinking strategies [14], and scrupulous pre-efficacy evaluation prior to pilot and efficacy testing. During Stage 1 (Steps 1–3), developers gather and synthesize insights from potential users and health behavior theory. In Stage 2 (Steps 4–7), developers iteratively and rapidly design and build the program while incorporating user feedback. Stage 3 (Steps 8–9) consists of rigorous evaluation of the program through pilot and efficacy testing. Stage 4 (Step 10) implores developers to hare their program and findings [13].

Since the dissemination of the IDEAS framework in 2016 [13], researchers have begun to explore the scope and applicability of this framework in different contexts and have detailed their experiences to varying degrees (e.g., Fedele et al. [15]; Kazemi et al. [16]; Liu et al. [17]). In these applications of the theory, researchers built new e-health behavioral interventions from inception to testing, but the question remained how the framework could be applied to guide a user-centered process of adapting an existing e-health intervention for use with a new priority population. Burchert et al. [18] explored this application for an e-health mental health intervention with Syrian refugees but only focused on Steps 4–6, excluding Steps 1–3, which emphasize *integrating insights from users and theory*. As such, we sought to answer the following research question, how can the IDEAS framework (Steps 1–7) guide the adaptation of an e-health physical activity intervention for use with a new priority population? The purpose of this manuscript is to describe the application of the IDEAS framework (Steps 1–7) in adapting a web-based physical activity intervention for use with young adult college students. 

## 2. Materials and Methods

A 10-week, web-based physical activity intervention originally developed for adults by Liu et al. [17] was selected for adaptation because of its rigorous developmental process, inclusion of health behavior change techniques (e.g., goal setting, self-monitoring, modifying physical environment) [17], and foundation in the Multi-Process Action Control framework (M-PAC), a framework that addresses the formation of motivation while emphasizing the gap between intention and physical activity behavior [19]. 

We adapted the original web-based intervention [17] for use with young adult college students. The adaptation process was guided by Steps 1–7 of the IDEAS framework and included two rounds of qualitative interviews during Steps 1 and 6. This project was conducted from June to December of 2020 by an interdisciplinary team of researchers and graduate students with expertise in nursing, health communication and marketing, digital communication technology, exercise psychology, and community health education. 

### 2.1. Step 1: Empathize with Target Users 

The first step was to learn about the needs, values, and preferences of target users via brief questionnaires, focus groups, and semi-structured interviews. A purposive sample was recruited from a large, urban university in the Midwestern USA. The inclusion criteria were: (1) ages 18–25 years, (2) undergraduate student, and (3) less than 150 min of moderate-to-vigorous physical activity per week. Ten target users expressed interest in participating in the study. Seven participated in a focus group or interview (Table 1), while three participants decided not to participate for reasons not disclosed. Participants received a $25 prepaid gift card. 

Study sessions were conducted virtually and lasted 1–2 h. Participants reviewed two lessons from the original web-based intervention and then engaged in a semi-structured focus group or interview led by the first two authors (K.R.H., L.J.D.). These authors are doctorally prepared (PhD) in nursing (K.R.H.) and public health with expertise in intercultural health communication and mixed-methods research with training in utilization evaluation (L.J.D.). The authors followed an interview protocol to lead the focus groups and interviews. The interview questions (e.g., What factors made it easy or difficult to get around the website and complete the lessons?) were grounded in areas of intervention feasibility defined by Bowen et al. [20]. Participants were invited to share their experience with the intervention. Questions were designed to help elicit participants’ specific thoughts about the following elements of the web-based physical activity intervention: (1) desired features; (2) attitudes towards the program including perceived satisfaction, acceptability, and appropriateness of the educational content; (3) perceived desire for and usefulness of the program; and (4) factors that may affect target user ability to complete the program. Participants were also asked to report the time for lesson completion and number of weeks they would be willing to commit to the physical activity intervention. At the end of each session, the authors summarized and recapped ideas and comments with the participants as an initial member validation check.

Sessions were audio-recorded and transcribed verbatim to ensure accuracy of data. We used Microsoft Word and QDA Miner to manage and inductively code the qualitative data. Thematic analysis procedures were inspired by Braun and Clarke [21]. After the focus groups concluded, the authors reread their field notes and then met to debrief and engage in collective conversation about initial broad ideas emerging from the data. Using latent feasibility concepts (e.g., acceptability, practicality) as a starting point [20] to frame the analysis, the lead author (K.R.H.) and second author (L.J.D.) then independently read the interview transcripts to further familiarize themselves with the data. They allowed semantic codes derived from the data to emerge in an effort to capture the essence of interview responses. These semantic codes were briefly discussed, and broad patterns and thematic categories were identified based on recurrence, repetition, and forcefulness [22]. The authors then continued coding and categorizing the data. As coding progressed, the authors re-conceptualized original themes and patterns into three metathemes (i.e., engagement, accountability, and cultural fit). Additionally, sub-themes were identified to define and explicate the metathemes [21]. The thematic framework eventually produced no new insights, suggesting that the data approached theoretical saturation despite the small sample size. Finally, the first two authors met to discuss and interpret the results. As an investigative triangulation check, the third author (K.M.K.) reviewed the findings and provided feedback on the interpretation of the data. The findings were also shared with participants in step 6, which served as a member check [23].

### 2.2. Step 2: Specify Target Behavior

The aim of Step 2 was to refine the target behavior to be specific, measurable, and consistent with current evidence and user preferences for this priority population [13]. The team reviewed current public health guidelines [24], empirical evidence (e.g., Saint-Maurice et al. [25]; Salin et al. [26]), and target user preferences from Step 1. For example, target users who participated in Step 1 discussed the importance of personalization of program features and physical activities they enjoyed including walking/jogging, yoga, fitness videos, spikeball, and gym workouts. Contextual factors such as safety and access were also considered to ensure the achievability of the target behavior. For example, public health mandates during the COVID-19 pandemic affected physical activity access by restricting gym entry and group fitness options [27].

### 2.3. Step 3: Ground in Behavioral Theory

The goal of Step 3 was to gain knowledge and understanding of health behavior theory and its role in health education programs. The strategies, concepts, and desired features generated during Steps 1 and 2 were evaluated for fit within the M-PAC framework [19] and the Behavior Change Technique Taxonomy of Michie et al. [28]. 

### 2.4. Step 4: Ideate Implementation Strategies

The team synthesized the data and information gathered from Steps 1 to 3 and developed creative solutions to modify the original program. The team met regularly to discuss design features and content modifications. At the end of each brainstorming session, ideas were prioritized into three categories: (1) Yes—strategy likely effective and feasible, (2) Maybe—strategy needs further investigation, and (3) Future consideration—strategy not currently feasible (Table 2). Notes and conclusions were circulated among the team for review and comment. 

### 2.5. Step 5: Prototype Potential Product 

The iterative process continued as the research team and the information technology (IT) development team discussed the feasibility of implementing specific design ideas that germinated during Step 4. Thorough documentation and regular collaboration between the research and IT development teams ensured that new design features aligned with focus group findings, theory, empirical evidence, and capabilities of the web-based platform. The modifications to content, design, and features were made using a step-by-step process outlined by the IT development team to ensure accuracy and completeness of modifications.

### 2.6. Step 6: Gather User Feedback on the Prototype

During Step 6, target users from the focus groups in Step 1 were asked to participate in individual semi-structured interviews led by the lead author (K.R.H.) to review the team’s interpretations and application of the findings from Step 1 and provide input on the prototype. Three of the initial seven participants elected to participate in the follow-up virtual interviews, which were recorded for accuracy. The author followed a semi-structured interview protocol to lead the interviews. The interview questions were designed to elicit discussion about whether the identified themes and desired features accurately reflected the focus group discussions from Step 1 and if the program modifications addressed the desired features. The participants had the opportunity to clarify inaccuracies and make additional suggestions for improvement. When a suggested modification was not feasible, an explanation with an alternative solution was discussed and additional feedback was requested. For example, one suggestion from Step 1 was for interactive diagrams; however, the web-based platform lacked this functionality. Instead, the ratio of interactive to non-interactive content was increased using other features (e.g., reflective questions, knowledge-check quizzes). The suggested idea was then placed in the Future Considerations category. At the end of each interview, the lead author summarized and recapped ideas and comments with the participant to ensure accuracy. Participants received a $20 prepaid gift card. Qualitative data were analyzed based on recurrence, repetition, and forcefulness [22] by the lead author and coded by hand using Microsoft Word.

### 2.7. Step 7: Build Minimum Viable Product

The aim of Step 7 was to build a fully functional but minimally viable beta version of the new physical activity intervention for pilot testing. Modifications were repeatedly reviewed to ensure accuracy and adherence to prior steps of the development process. Functionality was iteratively tested by the research and IT development teams. Emphasis was placed on creating a smooth user experience including ease of interaction with content through a simple consistent layout, interactivity for users within each lesson, multimedia content presentation, and branding of the site to include the university colors and identity elements for cohesiveness. Step 7 also included ensuring adequate functionality for pilot testing; for example, procedures and capabilities for participant enrollment and program evaluation using web analytics, along with reliable and validated process and outcome measures. 

## 3. Results

### 3.1. Step 1: Empathize with Target Users

Three metathemes were derived from the data during Step 1: oEngagement: fostering ongoing engagement with the program;oAccountability: need for personal and social accountability;oCultural fit: cultural fit within college lifestyle is imperative.

The first metatheme, *engagement*, encompassed several subthemes to foster ongoing interaction with the program. For example, participants desired delivery of educational content via mixed-media modalities (i.e., videos, graphs). Almost all participants preferred a combination of narrative reading, bulleted lists, infographics, whiteboard videos, and other brief videos to keep the lessons interesting while concisely delivering the content. They also found visual chunking of material and eye-catching graphics and formatting more engaging than plain text. For example, participant 6 stated *“… when there’s color and there’s like, you know, a bunch of different little things here and there, like, it doesn’t look as much of like, it’s more, I don’t know, it just doesn’t look much like work. Like, it looks more fun.”* The second subtheme to support engagement was the use of interactive content including interactive buttons and diagrams, reflective questions, and short knowledge-check quizzes. Participants also requested additional physical activity ideas and “how-to” instructions for physical activity as a feature. The fourth subtheme within the engagement metatheme was social connection. Participants preferred to exercise with friends, family, or pets, and expressed a desire for a digital social platform to provide social connection and support for physical activity. For example, participant 2 stated, *“I never go out on walks alone. I always take my nephew or my mom and like everybody because I can’t go alone.”*

The second metatheme was a need for personal and social *accountability*. There was consensus that following through with intentions to be physically active was difficult and having accountability built into the program would be helpful. Most participants desired the ability to track and evaluate personal progress towards a goal. Participant 1 suggested, *“being able to check something off the next week that you go on to it and say, ‘OK, I did this much,’ and you can track your progress as well so that way they can see how well they’re doing or what they’re not*.” There was debate about whether peer or coach accountability check-ins would be helpful; however, the participants overwhelmingly thought built-in reminders to be physically active would help them stay on track. There were mixed responses to using competition via social media for accountability. Participant 2 stated that social media could increase social accountability, *“It’s just like when nobody knows that you’re doing something, you have a higher chance of just dropping it because no one knows. But if people know about it, then you get more motivated to like do it with them.”* But *participant 4* pointed out that they were *“definitely too scared to go on the app to talk to an instructor or coach, so I definitely would rather have like a calendar or available classes at the [student recreation center]”.*

Finally, *cultural fit* within the college lifestyle emerged as the third metatheme. Specifically, discussions centered on the importance of content relevance and optimal lesson delivery. The participants commented that they could tell the program was not created specifically for them. *“I don’t think this was like specifically geared toward students*”, participant 4 pointed out. The participants desired topics that were more relevant to their life stage, specifically stress management, mental health, and well-being. *Participant 5* suggested, *“I think specifically the stress but also depression and anxiety, I think everyone knows someone who has it or has it themselves, I feel like, so that’s like something that everyone is very like aware of right now.”* The participants emphasized that the program should fit within their daily schedule and that features should be individualized based on user preferences. The lesson length (10–15 min) was deemed appropriate; however, participants preferred that the lessons be condensed with less repetition of information. Lesson summaries were helpful. For example, participant 5 pointed out that lesson summaries were *“nice because sometimes you get a lot of information and then you don’t really know what to do with it, so the end was nice to like just be like, this is what you learned in case you forgot.”* The participants agreed that one lesson per week would fit their schedules and a program completion date prior to finals would be appreciated. They also noted that competing obligations during the semester may interfere with completing the program. In general, participants reported that the original 10-week physical activity program was too long and that a 6- to 8-week program was more reasonable. This was supported by quantitative data gathered from the post-lesson review questionnaires in which participants were willing to commit to a program of 6.7 weeks in length (*N* = 7; *M* [SD] = 6.7 [2.7]; range = 2–10). 

### 3.2. Step 2: Specify Target Behavior

The target behavior was refined to include: (1) 150 min of moderate-intensity aerobic physical activity per week [24], and/or (2) increased steps per day as specified by the individual based on their current activity level. This would allow users to personalize their physical activity goals based on individual preferences, the types of physical activity they enjoy, and current activity level.

### 3.3. Step 3: Ground in Behavioral Theory

The themes (e.g., accountability) and desired features (e.g., self-monitoring, social support) generated during Steps 1 and 2 were consistent with the M-PAC framework [19] and the Behavior Change Technique Taxonomy [28]. Because the content and behavior change techniques used in the original program [17] were grounded in the M-PAC framework, the major lesson topics and techniques remained similar in the adapted program, including (1) Benefits of physical activity, (2) Self-confidence for physical activity, (3) Emotional regulation, (4) Building social physical activity opportunity and social support, (5) Goal setting and planning, (6) Self-monitoring, (7) Forming a habit, and (8) Physical activity identity.

### 3.4. Steps 4 and 5: Ideate Implementation Strategies and Prototype Potential Product

Steps 4 and 5 occurred in a highly iterative process as the team synthesized the data gathered from previous steps, developed creative solutions to modify the original program, and worked with the IT development team to assess the ideas for feasibility given resources and platform functionality. When a constraint was identified due to limited resources, functionality of the web-based platform, or another external entity (e.g., legal or IT security), the research and IT development teams revisited and revised the solutions and strategies. For instance, due to budgetary constraints, existing free technology (e.g., social media platforms) was used to create social connection rather than developing new technology. Table 3 outlines the program modifications corresponding to each theme.

### 3.5. Step 6: Gather User Feedback on the Prototype 

Three target users from Step 1 participated in the Step 6 interviews to validate resulting themes and corresponding program modifications. Feedback was positive. Participants agreed that the identified themes and modifications accurately reflected the perspectives presented during the focus groups in Step 1. No new themes emerged during Step 6 interviews, which further supported the initial thematic structure reached during Step 1. Three points of clarification were made during these interviews. First, participants suggested that the wellness coach be available to assist with the creation of goals and to provide additional education content on how to exercise (Metatheme: Fostering ongoing engagement with the program). Second, participants desired a digital space to share achievements and see each other’s progress (Metatheme: Need for personal and social accountability). Lastly, a combination of mobile application and website access was preferred over singular access (Metatheme: Cultural fit); however, this last suggestion was set aside for future consideration due to budgetary constraints. Given these new data, the research team repeated Steps 4 and 5. The wellness coach role was refined to include education on how to do physical activity and assist with goal creation. The purpose of the social media group was revised to include sharing of achievements and progress towards goals (Table 3).

### 3.6. Step 7: Build Minimum Viable Product

Through this rigorous redesign process that included repeat qualitative interviews, a markedly different health behavior intervention emerged. The newly adapted 8-week web-based physical activity intervention consisted of weekly interactive lessons, an expanded resource library with local resources and “how-to” videos, manual entry step and goal trackers, and a private social media group to be led by a wellness coach. Minimal modifications to the theory-based lesson content and behavior change techniques were needed. Instead, the major modifications to the intervention centered around strategies to foster ongoing engagement with the program, support accountability, and enhance cultural fit within the college student lifestyle. For example, the newly adapted intervention had a shorter duration (8 weeks) and more relevant graphics, examples, and statistics for the new priority population. Furthermore, the narrative text was shorter and the content related to prevention of chronic illness was reduced, while the content on the effects of physical activity on mental health and mood were emphasized. Figure 2 shows a sample of the web-based design.

The newly adapted intervention was designed specifically for young adult college students, supported by empirical evidence, and grounded in health behavior theory. Upon completion of Step 7, the newly adapted program was ready for pilot testing and included all functionality needed to evaluate the program such as links to online questionnaires and end-user data analytic capabilities to measure ongoing program engagement (e.g., time spent per lesson, lesson access per week, resource/page usage per participant, program access over intervention duration).

## 4. Discussion

This manuscript described a case example of the application of the IDEAS framework to guide the process of adapting an existing web-based physical activity intervention for a new priority population, specifically young adult college students. Using the IDEAS framework allowed for a progressive yet iterative design and development process that ensured that appropriate attention and consideration were given to the critical steps of adapting an e-health behavior change intervention for a new priority population. The qualitative results from Steps 1 and 6 provided important insights into the lives, needs, and preferences of the priority population. These insights prevented the team from making inaccurate assumptions about the target users and allowed for corrections to be made during the design phase rather than at later and potentially more costly stages of pilot and efficacy testing. It was particularly valuable to obtain the second round of feedback from the target users during Step 6 about the intervention modifications and prototype. As a result, we were able to make additional changes to the intervention design prior to the build stage (Step 7).

A few of the desired features identified during the focus groups and interviews were consistent with previous e-health research with young adults such as a desire for simple, structured interfaces with customizable features that can be tailored to personal preferences [29,30], the usefulness of timely reminders [7,9,30], and the desire for concurrent access to the content via website and mobile applications. However, support for the use of a digital platform for social connection was less clear. Results from previous research suggest that young adults are hesitant to post health behavior information on social media and their desire for competition is mixed [8,30]. Young adults may, however, be amenable to posting such information via a private social media group among people with similar goals [8]. Participants in this study appeared more open to the idea.

Lastly, the iterative process allowed for quick modifications during the design and development phases based on external factors beyond the team’s control. For example, the university’s IT security and legal departments identified concerns about incorporating some of the desired technology. Negotiations among the included parties caused minor delays in the process; however, the team was able to move forward with the design and development of the program with a plan to incorporate additional technology at a later stage. 

The next step for this project will be pilot testing (Step 8), including preliminary efficacy and thorough usability testing with user data analytics. Then, the project will progress to Step 9, which will include efficacy testing using a randomized controlled trial. Step 10 will be dissemination of the program and results. 

Weaknesses of this project included the small number of participants during Steps 1 and 6, which may have limited the breadth available from the data and the depth of data collected within each subtheme. Despite robust recruitment efforts, low participation was thought to be due to recruitment occurring during the summer when fewer students were enrolled in classes and at a time of significant historical events in the community that directly affected the student population such as social unrest, daily protests, and the early stages of the COVID-19 pandemic. Similarly, the methodology of this process could have been enhanced through the use of quantitative methods with larger samples. Furthermore, it is acknowledged that the target users reviewing sample lessons may have different experiences and behaviors than those who will complete the full program. Therefore, additional data on acceptability, demand, and practicality will be collected during pilot testing with a larger sample. In future studies, legal and IT security experts will be consulted earlier during Step 4 (Ideate implementation strategies) to avoid unnecessary delays later in the process. Lastly, as this was a case example with limited generalizability, the usefulness of this framework for other health behaviors and populations should continue to be explored.

## 5. Conclusions

This manuscript described the application of the IDEAS framework (Steps 1–7) to guide the adaptation of an existing web-based physical activity intervention for a new priority population, young adult college students. Modifications to the intervention centered around engagement, accountability, and cultural fit. An interdisciplinary team of developers and researchers collaborating with the priority population was crucial to conducting the strategic, user-centered adaptation process [13]. The development and adaptation of e-health behavior change interventions for new populations can be time- and resource-intensive. The IDEAS framework provided an efficient user-centered process for integrating evidence, behavior change theory, and user preferences and feedback into adapting the e-health behavior change intervention for a new population. Use of the iterative and user-centered framework allowed us to identify and address barriers to acceptability during the design and build stages rather than at later stages of pilot and efficacy testing. Researchers and practitioners should continue to explore the applicability of this framework for the development and adaptation processes of other e-health behavior interventions and populations.

## Figures and Tables

**Figure 1 healthcare-10-00700-f001:**
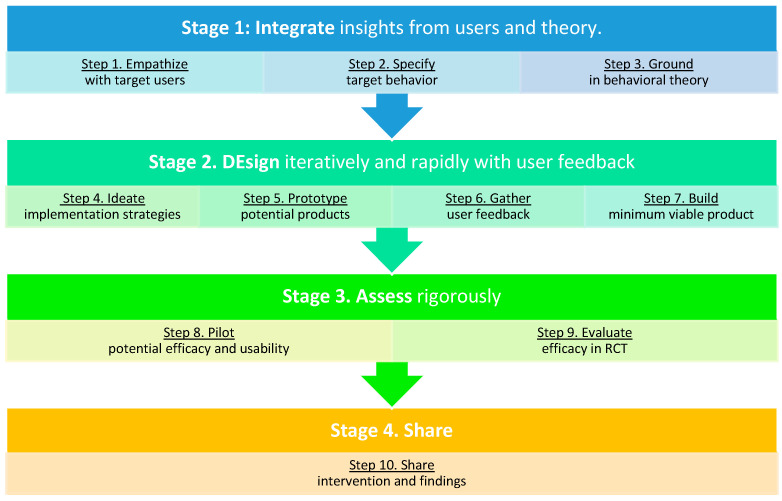
Adapted from IDEAS (*Integrate*, *DEsign*, *Assess*, and *Share*) framework by Mummah et al. [13].

**Figure 2 healthcare-10-00700-f002:**
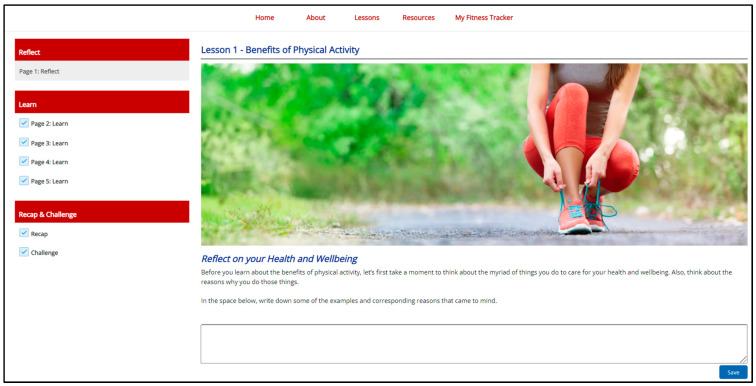
Sample lesson showing web design.

**Table 1 healthcare-10-00700-t001:** Demographic characteristics of Step 1 sample (*N* = 7).

Characteristic	*n*
**Age**	
20	2
21	3
22	0
23	2
**Gender**	
Female	5
Genderqueer	1
Male	1
**Race and/or Ethnicity**	
African	1
American Indian, Alaskan Native or Native Hawaiian	1
Asian or Pacific Islander	1
Black or African American	2
White	4

**Table 2 healthcare-10-00700-t002:** Step 4 prioritization of implementation strategies—example.

Yes	Maybe	Future Consideration
Decrease narrative text	Incorporate social media platform for social connection	Add interactive diagrams and buttons
Increase use of brief videos, infographics, pictures, graphs	Manual entry activity tracker	Add personalized avatar and gamification of progress
University branding for cohesiveness	Have university students create inclusive physical activity videos using local resources	Integrate with personal fitness trackers
Modify contents for relevance to college lifestyle	Wellness coach	

**Table 3 healthcare-10-00700-t003:** Themes, supportive quotations, and program modifications.

Themes with Supportive Target User Quotations	Modifications to Program (Affected Lessons & Program Components)
**Metatheme 1: Fostering Ongoing Engagement with Program**	
(1) Mixed media *“I am not really big on like reading a whole lot … I liked the little text boxes that had like, ‘Did you know’ in it. That, for me, I like to read stuff like that because it’s attention grabbing and it’s short and just quick and to the point and the little info-graph, kind of poster-looking thing, I liked that, and the video, I liked the video a lot too.” (Participant 7)*	Decreased narrative content (Lessons 1–8) Increased use of pictures, short videos, bullet points, infographics, graphs, etc. (Lessons 1–8)
(2) Interactive content *“I liked the way it’s kind of interactive rather than just reading because I can’t just sit and read. It will just go in one ear and out the other, kind of.” (Participant 7)*	Increased amount of interaction required to progress through lessons (Lessons 1–8) Increased ratio of interactive content to non-interactive content (Lessons 1–8)
(3) Physical activity ideas and how-to instructions *“But like the first step is usually the hardest. So, I think, yeah, like having a section or whatever you’re thinking about to like have options or like instructions or suggestions for how to like start doing something would be really good.” (Participant 5)*	Added more “how-to” videos (Resources library) Added physical activity ideas for various contexts (indoor/outdoor, group/individual) (Lessons 4; Resources library) Added local resources (Resources library) Added a wellness coach to work with participants via social media (Social media group; Lessons 1–8) Wellness coach to assist with creation of goals and exercise instructions, if desired (Social media group; Lessons 1–8) *
(4) Social connection *“… and possibly like ways to do it with friends or people that you know because like college students are always trying to find like ways to hang out with people and stuff, so maybe like bringing in the social aspect of physical activity?” (Participant 5)*	Created private social media group led by a wellness coach (Social media group)
**Metatheme 2: Need for personal and social accountability**	
(1) Personal and social accountability *“I think [trackers are] a great idea to help you get motivated to see how you’re doing and keep coming back… then like weekly charts so you can see your progress, if you are doing good one week or one day you’re doing good, it shows you like progress and graphs.” (Participant 2)*	Added manual entry steps tracker graph (Trackers) Added manual entry goal tracker calendar (Trackers) Wellness coach and social media group added for accountability (Social media group; Lessons 1–8) Purpose of social media group clarified to include the sharing of achievements and progress (Social media group) *
**Metatheme 3: Cultural fit within college lifestyle is imperative**	
(1) Relevance *“I think mental health would be biggest for us just because I think our generation feels, like I was talking to someone the other day and I think the reason like with COVID and traumatic things that happen, … because ever since we were born it was like 9/11, school shootings, blah, blah, blah, so I think that mental health is a big deal in our generation …” (Participant 6)*	Updated content, images, and statistics to be relevant to student life (Lessons 1–8) Deleted content deemed unimportant by participants Increased emphasis on content related to mental health, connectedness, and well-being (Lessons 1, 2, and 3) Increased feature options for personalization of program (Trackers; Social media group; Wellness coach; Lessons 1–8)
(2) Lesson delivery *“… unless I’m looking back at it in my planner or somebody reminds me or texts me … I won’t think twice about it and I’ll be like, ‘Oh man, I didn’t do it. Catch it next time.’” (Participant 1)*	Maintained lesson length (10–15 min) and frequency (once per week) (Lessons 1–8) Added semi-weekly reminders (Lessons 1–8; Notifications) Decreased program length from 10 to 8 weeks (Lessons 1 and 4) Increased use of bolding, headers, and colors for organization and readability (Lessons 1–8)

Note: * Revision made to program during Step 6: Gather user feedback on the prototype.

## Data Availability

All data presented in this study are available upon request from the corresponding author.

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
