# Peer review of "Application of the IDEAS Framework in Adapting a Web-Based Physical Activity Intervention for Young Adult College Students"

_healthcare, 2022, doi:10.3390/healthcare10040700_

Round 1
Reviewer 1 Report
The authors must be commended on their application of the ideas framework to adapt and refine a web-based physical activity behaviour change intervention for college students in the US. The paper is well-written and easy to follow, with minimal if any grammatical or spelling errors. The strengths of this paper is a very well structured and transparent approach to detailing the data collection, and translating the qualitative feedback into actionable items to make the resultant intervention more likely to be appropriate for this college population. Weaknesses, as identified by the authors, are the small sample size which was put down to a combination of semester timing, protests and COVID-19 disruptions. I have only a few suggestions below for the authors, and look forward to seeing the results of the pilot study of this intervention.
Comments:
- The introduction should have at least a brief overview of current physical activity habits of American children in terms of prevalence achieving US guidelines, minutes of activity, etc to compliment their section about the declining PA habits in adolescents
- In Table 2, given the small sample size I do not think the provision of % for each category is appropriate or required. Please remove.
- After reviewing the original intervention by Liu et al 2015, of which this intervention was adapted, it would be helpful to have as an appendix (if permitted) of a table similar to that of table 3 in Liu et al 2015 that outlines the week-by-week program that was developed at the end of this IDEAS process for the 8-week PA web-based program.
Overall, well done on the design, implementation and write up of this study.
Author Response
Thank you for your thoughtful comments and feedback to improve the manuscript. We hope we have adequately addressed any questions and concerns. Please find itemized responses to the reviewer comments in the attached document. Thank you.

Reviewer 2 Report
The opinions discussed in this article whose title is “Application of the IDEAS Framework in Adapting a Web-based Physical Activity Intervention for Young Adult College Students” are innovative and meaningful to practice. But there are still some issues that need to be fixed further. Details are as follows.
â‘ 1. Introduction
- It is recommended to arrange the order of the contents of "Introduction" to make them more organized to read.
- As an intervention study, the contents of "Research Hypothesis" and "Research Implications" are suggested to be added after the research purpose.
â‘¡ 2. Materials and Methods
Please refer to the "SRQR" and "COREQ", the reporting guidelines for qualitative studies, to supplement the current missing research contents.
- 2.1. Step 1: Empathize with Target Users
- How many participants were in the study?How many people refused to participate or dropped out? Reasons?
- Were questions, prompts, guides provided by the authors? Was it pilot tested?
- 2.6. Step 6: Gather User Feedback on the Prototype
- Please add relevant information about the interviewers/facilitators: (1) which author/s conducted the interview or focus group? (2) what were the researcher’s credentials? (3) what experience or training did the researchers have?
- Please subjoin the contents about the "data analysis" after obtaining the qualitative interview data. For example the following: (1) How many data coders coded the data? (2) The description of the coding tree? (3) What software, if applicable, was used to manage the data?
â‘¢ 3. Results
- 3.1. Step 1: Empathize with Target Users
- Only 7 college students were recruited to participate in qualitative interviews. Was data saturation discussed?
- Were transcripts returned to participants for comment and/or correctionafter interview guide?
- Were 3 themes identified in advance or derived from the data?
- The “Results”section may be preferably written in bulleted points if possible to make it easy for the reader to understand.
- 3.6. Step 7: Build Minimum Viable Product
You can explain in detail how this website implements interventions for college students, including intervention content, data collection, monitoring methods, etc.
Author Response
Thank you for your thoughtful comments and suggestions to improve the manuscript. Please find responses for each comment in the attached document. We hope we have adequately addressed all comments and concerns. Thank you.

Reviewer 3 Report
Dear Author, I have read your manuscript and forwarded a few comments.
Here attached, please find my comments for the improvement of your manuscript.
Thanks!

Author Response
Thank you for your thoughtful comments and helpful suggestions to improve the manuscript. Please see responses to each comment in the attached document. We hope we have adequately addressed any comments and concerns. Thank you.

Round 2
Reviewer 2 Report
Dear Editor
Thanks for inviting me again to evaluate the revised version of manuscript healthcare-1658775 entitled "Application of the IDEAS Framework in Adapting a Web-based Physical Activity Intervention for Young Adult College Students". The revised paper is well-written and is acceptable for being published.